# Early Stopping for Nonparametric Testing

**Meimei Liu**
Department of Statistical Science
Duke University
Durham, NC 27705
`meimei.liu@duke.edu`

**Guang Cheng**
Department of Statistics
Purdue University
West Lafayette, IN 47907
`chengg@purdue.edu`

## Abstract

Early stopping of iterative algorithms is an algorithmic regularization method to avoid over-fitting in estimation and classification. In this paper, we show that early stopping can also be applied to obtain the minimax optimal testing in a general non-parametric setup. Specifically, a Wald-type test statistic is obtained based on an iterated estimate produced by functional gradient descent algorithms in a reproducing kernel Hilbert space. A notable contribution is to establish a "sharp" stopping rule: when the number of iterations achieves an optimal order, testing optimality is achievable; otherwise, testing optimality becomes impossible. As a by-product, a similar sharpness result is also derived for minimax optimal estimation under early stopping. All obtained results hold for various kernel classes, including Sobolev smoothness classes and Gaussian kernel classes.

## 1 Introduction

As a computationally efficient approach, early stopping often works by terminating an iterative algorithm on a pre-specified number of steps to avoid over-fitting. Recently, various forms of early stopping have been proposed in estimation and classification. Examples include boosting algorithms (Bühlmann and Yu [2003], Zhang and Yu [2005], Wei et al. [2017]); gradient descent over reproducing kernel Hilbert spaces (Yao et al. [2007], Raskutti et al. [2014]) and reference therein. However, *statistical inference* based on early stopping has largely remained unexplored.

In this paper, we apply the early stopping regularization to nonparametric testing and characterize its minimax optimality from an algorithmic perspective. Notably, it differs from the traditional framework of using penalization methods to conduct statistical inference. Recall that classical nonparametric inference often involves minimizing objective functions in the *loss + penalty* form to avoid overfitting; examples include the penalized likelihood ratio test, Wald-type test, see Fan and Jiang [2007], Shang and Cheng [2013], Liu et al. [2018] and reference therein. However, solving a quadratic program in the penalized regularization requires $O(n^3)$ basic operations. Additionally, in practice cross validation method (Golub et al. [1979]) is often used as a tuning procedure which is known to be optimal for estimation but suboptimal for testing; see Fan et al. [2001]. As far as we are aware, there is no theoretically justified tuning procedure for obtaining optimal testing in our setup. We address this issue by proposing a data-dependent early stopping rule that enjoys both theoretical support and computational efficiency.

To be more specific, we first develop a Wald-type test statistic $D_{n,t}$ based on the iterated estimator $f_t$ with $t$ being the number of iterations. As illustrated in Figure 1 (a) and (b), the testing power demonstrates a parabolic pattern. Specifically, it increases as the iteration grows in the beginning, and then decreases after reaching its largest value when $t = T^*$, implying how over-fitting affects the power performance. To precisely quantify $T^*$, we analyze the power performance by characterizing the strength of the weakest detectable signals (SWDS). We show that SWDS at each iteration is controlled by the bias of the iterated estimator and the standard derivation of the test statistic. In

fact, each iterative step reduces the former but increase the latter. Such a tradeoff in testing is rather different from the classical "bias-variance" tradeoff in estimation; as shown in Figure 1 (c). Hence, the early stopping rule to be provided is different from those in the literature such as Raskutti et al. [2014] and Wei et al. [2017]; also see Figure 1 (a) and (b) in comparison with power and MSE.

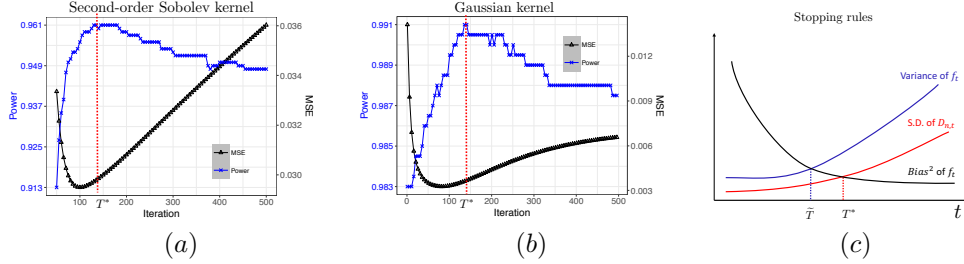

$(a)$          $(b)$          $(c)$

Figure 1: $(a)$ and $(b)$ are mean square error (MSE) and power performance of gradient descent update at each iteration with constant step size $\alpha = 1$; Power was calculated based on 500 replicates. $(a)$ Data were generated via $y_i = 0.5x_i^2 + 0.5\sin(4\pi x_i) + \epsilon_i$ with sample size $n = 200$, $\{x_i\}_{i=1}^n \sim Unif[0,1]$, $\epsilon_i \sim N(0,1)$. $(b)$ Data were generated by $y_i = 0.5x_i^2 + 0.5|x_i - 0.5| + \epsilon_i$ with sample size $n = 200$. $(c)$ Stopping rules for estimation and testing based on different tradeoff criteria.

The above analysis apply to many reproducing kernels, and lead to specific optimal testing rate, depending on their eigendecay rate. In the specific examples of polynomial decaying kernel and exponential decaying kernel, we further show that the developed stopping rule is indeed "sharp": testing optimality is obtained if and only if the number of iterations obtains an optimal order defined by the stopping rule. As a by-product, we prove that the early stopping rule in Raskutti et al. [2014] and Wei et al. [2017] is also "sharp" for optimal estimation.

## 2    Background and Problem Formulation

We begin by introducing some background on reproducing kernel Hilbert space (RKHS), and functional gradient descent algorithms in the RKHS, together with our nonparametric testing formulation.

### 2.1    Nonparametric estimation in reproducing kernel Hilbert spaces

Consider the following nonparametric model

$$y_i = f(x_i) + \epsilon_i, \quad i = 1, \cdots, n, \tag{2.1}$$

where $x_i \in \mathcal{X} \subset \mathbb{R}^d$ for a fixed $d \geq 1$ are random covariates, and $\epsilon_i$ are Gaussian random noise with mean zero and variance $\sigma^2$. Throughout we assume that $f \in \mathcal{H}$, where $\mathcal{H} \subset L^2(P_X)$ is a reproducing kernel Hilbert space (RKHS) associated with an inner product $\langle \cdot, \cdot \rangle_{\mathcal{H}}$ and a reproducing kernel function $K(\cdot, \cdot) : \mathcal{X} \times \mathcal{X} \to \mathbb{R}$. By Mercer's Theorem, $K$ has the following spectral expansion:

$$K(x, x') = \sum_{i=1}^{\infty} \mu_i \phi_i(x) \phi_i(x'), \quad x, x' \in \mathcal{X}, \tag{2.2}$$

where $\mu_1 \geq \mu_2 \geq \cdots \geq 0$ is a sequence of eigenvalues and $\{\phi_i\}_{i=1}^{\infty}$ form a basis in $L^2(P_X)$. Moreover, for any $i, j \in \mathbb{N}$,

$$\langle \phi_i, \phi_j \rangle_{L^2(P_X)} = \delta_{ij} \quad \text{and} \quad \langle \phi_i, \phi_j \rangle_{\mathcal{H}} = \delta_{ij}/\mu_i.$$

In the literature, e.g., Guo [2002] and Shang and Cheng [2013], it is common to assume that $\phi_j$'s are uniformly bounded. This is also assumed throughout this paper.

**Assumption A1.** *The eigenfunctions $\{\phi_k\}_{k=0}^{\infty}$ are uniformly bounded on $\mathcal{X}$, i.e., there exists a finite constant $c_K > 0$ such that*

$$\sup_{j \geq 1} \|\phi_j\|_{\sup} \leq c_K.$$

Two types of kernel are often considered in the nonparametric literature, depending on how fast its eigenvalues decay to zero. The first is that $\mu_i \asymp i^{-2m}$, leading to the so-called polynomial decay kernel (PDK) of order $m > 0$. For instance, an $m$-order Sobolev space is an RKHS with a PDK of order $m$; see Wahba [1990], and the trigonometric basis in periodic Sobolev space with PDK satisfies Assumption A1 trivially. The second is that $\mu_i \asymp \exp(-\beta i^p)$ for some constant $\beta, p > 0$, corresponding to the so-called exponential-polynomial decay kernel (EDK) of order $p > 0$; see Schölkopf et al. [1999]. In particular, for EDK of order two, an example is $K(x_1, x_2) = \exp(-(x_1 - x_2)^2/2)$. In the latter case, Assumption A1 holds according to Lu et al. [2016].

By representer theorem, any $f \in \mathcal{H}$ can be presented as

$$f(\cdot) = \frac{1}{\sqrt{n}} \sum_{i=1}^{n} w_i K(x_i, \cdot) + \xi(\cdot),$$

where $\xi \in \mathcal{H}$ and $\xi(\cdot) \perp \text{span}\{K(x_1, \cdot), \cdots, K(x_n, \cdot)\}$. Given $\boldsymbol{x} = (x_1, \cdots, x_n)$, define an empirical kernel $[\boldsymbol{K}]_{ij} = \frac{1}{n} K(x_i, x_j)$ and $\boldsymbol{f} = (f(x_1), \cdots, f(x_n))$, then $\boldsymbol{f} = \sqrt{n} \boldsymbol{K} w$, where $w = (w_1, \cdots, w_n)^\top \in \mathbb{R}^n$.

## 2.2 Gradient Descent Algorithms

Given the samples $\{(x_i, y_i)\}$, consider minimizing the least-square loss function

$$\mathcal{L}(f) := \frac{1}{2n} \sum_{i=1}^{n} (y_i - f(x_i))^2$$

over a Hilbert space $\mathcal{H}$. Note that by representer theorem, $f(x) = \langle f, K(x, \cdot) \rangle_{\mathcal{H}}$, then the gradient of $\mathcal{L}(f)$ is $\nabla \mathcal{L}(f) = n^{-1} \sum_{i=1}^{n} (f(x_i) - y_i) K(x_i, \cdot)$. Given $\boldsymbol{x} = (x_1, \cdots, x_n)$ and $\boldsymbol{y} = (y_1, \cdots, y_n)$, define $\boldsymbol{f}_t = (f_t(x_1), \cdots, f_t(x_n))$ for $t = 0, 1, \cdots$. Then straightforward calculation shows that the functional gradient descent algorithm generates a sequence of vectors $\{\boldsymbol{f}_t\}_{t=0}^{\infty}$ via the recursion

$$\boldsymbol{f}_{t+1} = \boldsymbol{f}_t - \alpha_t \boldsymbol{K}(\boldsymbol{f}_t - \boldsymbol{y}), \tag{2.3}$$

where $\{\alpha_t\}_{t=0}^{\infty}$ is the step sizes. Denote the total step size upto the $t$-th step as $\eta_t = \sum_{\tau=0}^{t-1} \alpha_\tau$. Consider the singular value decomposition $\boldsymbol{K} = U\Lambda U^\top$, where $UU^\top = I_n$ and $\Lambda = \text{diag}(\widehat{\mu}_1, \widehat{\mu}_2, \cdots, \widehat{\mu}_n)$ with $\widehat{\mu}_1 \geq \widehat{\mu}_2 \geq \cdots \geq \widehat{\mu}_n \geq 0$. We have the following assumption for the step sizes and $\eta_t$.

**Assumption A2.** *The step size $\{\alpha_t\}_{t=0}^{\infty}$ is non-increasing; for all $\tau = 0, 1, 2, \cdots$, $0 \leq \alpha_\tau \leq \min\{1, 1/\widehat{\mu}_1\}$. The total step size $\eta_t = \sum_{\tau=0}^{t-1} \alpha_\tau$ diverges as $t \to \infty$; for $0 \leq t_1 \ll t_2$ as $t_2 \to \infty$, $\eta_{t_1} \ll \eta_{t_2}$.*

Assumption A2 supposes the step size $\{\alpha_t\}_{t=0}^{\infty}$ to be bounded and non-increasing, but cannot decrease too fast as $t$ diverges. Many choices of step sizes satisfy Assumption A2. A trivial example is to choose a constant step size $\alpha_0 = \cdots = \alpha_t = \min\{1, 1/\widehat{\mu}_1\}$.

Define $\kappa_t = \text{argmin}\{j : \mu_j < \frac{1}{\eta_t}\} - 1$, we have the following assumption on the population eigenvalues through $\kappa_t$.

**Assumption A3.** *$\kappa_t$ diverges as $t \to \infty$.*

It is easy to check that Assumption A3 is satisfied in PDK and EDK introduced in Section 2.1.

## 2.3 Nonparametric testing

Our goal is to test whether the nonparametric function in (2.1) is equal to some known function. To be precise, we consider the nonparametric hypothesis testing problem

$$H_0 : f = f^* \quad \text{v.s. } H_1 : f \in \mathcal{H} \setminus \{f^*\},$$

where $f^*$ is a hypothesized function. For convenience, assume $f^* = 0$, i.e., we will test

$$H_0 : f = 0 \quad \text{vs.} \quad H_1 : f \in \mathcal{H} \setminus \{0\}. \tag{2.4}$$

In general, testing $f = f^*$ (for an arbitrary known $f^*$) is equivalent to testing $f_* \equiv f - f^* = 0$. So, (2.4) has no loss of generality. Based on the iterated estimator $\boldsymbol{f}_t$, we propose the following Wald-type test statistic:

$$D_{n,t} = \|\boldsymbol{f}_t\|_n^2, \tag{2.5}$$

where $\|\boldsymbol{f}_t\|_n^2 = \frac{1}{n}\sum_{i=1}^n f_t^2(x_i)$. In what follows, we will derive the null limit distribution of $D_{n,t}$, and explicitly show how the stopping time affects minimax optimality of testing.

## 3 Main Results

### 3.1 Stopping rule for nonparametric testing

Given a sequence of step size $\{\alpha_t\}_{t=0}^\infty$ satisfying Assumption A2, we first introduce the stopping rule as follows:

$$T^* := \arg\min\left\{ t \in \mathbb{N} \mid \frac{1}{\eta_t} < \frac{\sigma}{n}\sqrt{\sum_{i=1}^n \min\{1, \eta_t\widehat{\mu}_i\}} \right\}. \tag{3.1}$$

As will be clarified in Section 3.2, the intuition underlying the stopping rule (3.1) is that $\frac{1}{\eta_t}$ controls the bias of the iterated estimator $f_t$, which is a decrease function of $t$; $\frac{1}{n}\sqrt{\sum_{i=1}^n \min\{1, \eta_t\widehat{\mu}_i\}}$ is the standard deviation of the test statistic $D_{n,t}$ as an increasing function of $t$. The optimal stopping rule can be achieved by such a *bias-standard deviation* tradeoff. Recall that such a tradeoff in (3.1) for testing is different from another type of bias-variance tradeoff in estimation (see Raskutti et al. [2014], Wei et al. [2017]), thus leading to different optimal stopping time. In fact, as seen in Figure 1 ($c$), optimal estimation can be achieved at $\widetilde{T}$, which is earlier than than $T^*$. This is also empirically confirmed by Figure 1 ($a$) and ($b$) where minimum mean square error (MSE) can always be achieved earlier than the maximum power. Please see Section 4 for more discussions.

### 3.2 Minimax optimal testing

In this section, we first derive the null limit distribution of (standardized) $D_{n,t}$ as a standard Gaussian under mild conditions, that is, we only require the total step sizes $\eta_t$ goes to infinity.

Define a sequence of diagonal shrinkage matrices as $S^t = \prod_{\tau=0}^{t-1}(I_n - \alpha_\tau \Lambda)$. As stated in Raskutti et al. [2014], the matrix $S^t$ describes the extent of shrinkage towards the origin. By Assumption A2 that $0 \leq \alpha_t \leq \min\{1, 1/\widehat{\mu}_1\}$, $S^t$ is positive semidefinite.

**Theorem 3.1.** *Suppose Assumption A2, A3 are satisfied. Then under $H_0$, as $n \to \infty$ and $t \to \infty$, we have*

$$\frac{D_{n,t} - \mu_{n,t}}{\sigma_{n,t}} \xrightarrow{d} N(0,1).$$

*Here $\mu_{n,t} = \mathrm{E}_{H_0}[D_{n,t}|\boldsymbol{x}] = \frac{1}{n}\mathrm{tr}((I_n - S^t)^2)$ and $\sigma_{n,t}^2 = \mathrm{Var}_{H_0}[D_{n,t}|\boldsymbol{x}] = \frac{2}{n^2}\mathrm{tr}((I - S^t)^4)$.*

Then based on Theorem 3.1, we have the following testing rule at significance level $\alpha$:

$$\phi_{n,t} = I(|D_{n,t} - \mu_{n,t}| \geq z_{1-\alpha/2}\sigma_{n,t}),$$

where $z_{1-\alpha/2}$ is the $100 \times (1 - \alpha/2)$th percentile of standard normal distribution.

**Lemma 3.2.** $\mu_{n,t} \asymp \frac{1}{n}\sum_{i=1}^n \min\{1, \eta_t\widehat{\mu}_i\}$, *and* $\sigma_{n,t}^2 \asymp \frac{1}{n^2}\sum_{i=1}^n \min\{1, \eta_t\widehat{\mu}_i\}$.

Define the squared separation rate

$$d_{n,t}^2 = \frac{1}{\eta_t} + \sigma_{n,t} \asymp \frac{1}{\eta_t} + \frac{1}{n}\sqrt{\sum_{i=1}^n \min\{1, \eta_t\widehat{\mu}_i\}}.$$

The separation rate $d_{n,t}$ is used to measure the distance between the null hypothesis and a sequence of alternative hypotheses. The following Theorem 3.3 shows that, if the alternative signal $f$ is separated from zero by an order $d_{n,t}$, then the proposed test statistic $D_{n,t}$ asymptotically achieves high power

at the total step size $\eta_t$. To achieve the maximum power, we need to minimize $d_{n,t}$. Under the stopping rule (3.1), we can see that when $t = T^*$, the separation rate achieves its minimal value as $d_n^* := d_{n,T^*}$.

**Theorem 3.3.** $(a)$ *Suppose Assumption A2 and A3 are satisfied. For any $\varepsilon > 0$, there exist positive constants $C_\varepsilon$, $t_\varepsilon$ and $N_\varepsilon$ such that with probability greater than $1 - e^{-c\kappa_t}$,*

$$\inf_{t \geq t_\varepsilon} \inf_{n \geq N_\varepsilon} \inf_{\substack{f \in \mathcal{B} \\ \|f\|_n \geq C_\varepsilon d_{n,t}}} P_f(\phi_{n,t} = 1 | \boldsymbol{x}) \geq 1 - \varepsilon,$$

*where $c$ is a constant, $\mathcal{B} = \{f \in \mathcal{H} : \|f\|_{\mathcal{H}} \leq C\}$ for a constant $C$ and $P_f(\cdot)$ is the probability measure under $f$.*

$(b)$ *The separation rate $d_{n,t}$ achieves its minimal value as $d_n^* := d_{n,T^*}$.*

The general Theorem 3.3 implies the following concrete stopping rules under various kernel classes.

**Corollary 3.4.** *(PDK of order $m$) Suppose Assumption A2 holds and $m > 3/2$. Then at time $T^*$ with $\eta_{T^*} \asymp n^{4m/(4m+1)}$, for any $\varepsilon > 0$, there exist constants $C_\varepsilon$ and $N_\varepsilon$ such that, with probability greater than $1 - e^{-c_m n^{(2m-3)/(2m-1)}} - e^{-c_1 n^{2/(4m+1)}}$,*

$$\inf_{\substack{n \geq N_\varepsilon}} \inf_{\substack{f \in \mathcal{B} \\ \|f\|_n \geq C_\varepsilon n^{-\frac{2m}{4m+1}}}} P_f(\phi_{n,T^*} = 1 | \boldsymbol{x}) \geq 1 - \varepsilon,$$

*where $c_m$ is an absolute constant depending on $m$ only, $c_1$ is a constant.*

Note that the minimal separation rate $n^{-\frac{2m}{4m+1}}$ is minimax optimal according to (Ingster [1993]). Thus, $D_{n,T^*}$ is optimal when $\eta_{T^*} \asymp n^{4m/(4m+1)}$. Note that $\eta_{T^*} = \sum_{t=0}^{T^*-1} \alpha_t$, $T^* \asymp n^{4m/(4m+1)}$ when constant step sizes are chosen.

**Corollary 3.5.** *(EDK of order $p$) Suppose Assumption A2 holds and $p \geq 1$. Then at time $T^*$ with $\eta_{T^*} \asymp n(\log n)^{-1/(2p)}$, for any $\varepsilon > 0$, there exist constants $C_\varepsilon$ and $N_\varepsilon$ such that, with probability greater than $1 - e^{-c_{\beta,p} n(\log n)^{-2/p}} - e^{-c_1(\log n)^{1/p}}$,*

$$\inf_{\substack{n \geq N_\varepsilon}} \inf_{\substack{f \in \mathcal{B} \\ \|f\|_n \geq C_\varepsilon n^{-\frac{1}{2}}(\log n)^{\frac{1}{4p}}}} P_f(\phi_{n,T^*} = 1 | \boldsymbol{x}) \geq 1 - \varepsilon,$$

*where $c_{\beta,p}$ is an absolute constant depending on $\beta, p$.*

Note that the minimal separation rate $n^{-1/2}(\log n)^{1/(4p)}$ is proven to be minimax optimal in Corollary 1 of Wei and Wainwright [2017]. Hence, $D_{n,T^*}$ is optimal at the total step size $\eta_{T^*} \asymp n(\log n)^{-1/(2p)}$. When the step sizes are chosen as constants, then the corresponding $T^* \asymp n(\log n)^{-1/(2p)}$.

### 3.3 Sharpness of the stopping rule

Theorem 3.3 shows that optimal testing can be achieved when $t = T^*$. In the specific examples of PDK and EDK, Theorem 3.6 further shows that when $t \ll T^*$ or $t \gg T^*$, there exists a local alternative $f$ that is not detectable by $D_{n,t}$ even when it is separated from zero by $d_n^*$. In this case, the asymptotic testing power is actually smaller than $\alpha$. Hence, we claim that $T^*$ is *sharp* in the sense that testing optimality is obtained if and only if the total step size achieves the order of $\eta_{T^*}$. Given a sequence of step size $\{\alpha_t\}_{t=0}^\infty$ satisfying Assumption A2, we have the following results.

**Theorem 3.6.** *Suppose Assumption A2 holds, and $t \ll T^*$ or $t \gg T^*$. There exists a positive constant $C_1$ such that, with probability approaching 1,*

$$\limsup_{n \to \infty} \inf_{\substack{f \in \mathcal{B} \\ \|f\|_n \geq C_1 d_n^*}} P_f(\phi_{n,t} = 1 | \boldsymbol{x}) \leq \alpha.$$

In the proof, we construct the alternative function as $\sum_{i=1}^n K(x_i, \cdot) w_i$, with $w_i$ being defined in (A.8) and (A.9) for the two cases $t \ll T^*$ and $t \gg T^*$, respectively.

# 4 Sharpness of early stopping in nonparametric estimation

In this section, we review the existing early stopping rule for estimation, and further explore its "sharpness" property. In the literature, Raskutti et al. [2014] and Wei et al. [2017] proposed to use the fixed point of local empirical Rademacher complexity to define the stopping rule as follows

$$\widetilde{T} := \arg\min \left\{ t \in \mathbb{N} \mid \frac{1}{\eta_t} < \frac{\sigma}{n} \sum_{i=1}^{n} \min\{1, \eta_t \widehat{\mu}_i\} \right\}. \tag{4.1}$$

Given the above stopping rule, the following theorem holds where $f^*$ represents truth.

**Theorem 4.1** (Raskutti et al. [2014]). *Given the stopping time $\widetilde{T}$ defined by (4.1), there are universal positive constants $(c_1, c_2)$ such that the following events hold with probability at least $1 - c_1 \exp(-c_2 n/\eta_{\widetilde{T}})$:*

(a) *For all iterations $t = 1, 2, \cdots, \widetilde{T}$: $\|f_t - f^*\|_n^2 \leq \frac{4}{e\eta_t}$.*

(b) *At the iteration $\widetilde{T}$, $\|f_{\widetilde{T}} - f^*\|_n^2 \leq 12\frac{1}{\eta_{\widetilde{T}}}$.*

(c) *For all $t \gg \widetilde{T}$,*

$$\mathrm{E} \|f_t - f^*\|_n^2 \geq \frac{\sigma^2}{4} \Big( \frac{1}{n} \sum_{i=1}^{n} \min\{1, \widehat{\mu}_i \eta_t\} \Big) \gg \frac{\sigma^2}{\eta_{\widetilde{T}}}.$$

To show the sharpness of $\widetilde{T}$, it suffices to examine the upper bound in Theorem 4.1 (a). In particular, we prove a complementary lower bound result. Specifically, Theorem 4.2 implies that once $t \ll \widetilde{T}$, the rate optimality will break down for at least one true $f \in \mathcal{B}$ with high probability. Denote the stopping time $\widetilde{T}$ satisfying

$$\eta_{\widetilde{T}} \asymp \begin{cases} n^{2m/(2m+1)} & K \text{ is PDK of order } m, \\ n/(\log n)^{1/p} & K \text{ is EDK of order } p. \end{cases}$$

**Theorem 4.2.** (a) *(PDK of order $m$) Suppose Assumption A2 holds and $m > \frac{3}{2}$. For all $t \ll \widetilde{T}$, with probability approaching 1, it holds that*

$$\sup_{f \in \mathcal{B}} \|f_t - f^*\|_n^2 \geq \frac{c_m \sigma^2}{\eta_t} \gg \frac{1}{\eta_{\widetilde{T}}}.$$

(b) *(EDK of order $p$) Suppose Assumption A2 holds and $p \geq 1$. For all $t \ll \widetilde{T}$, with probability approaching 1,*

$$\sup_{f \in \mathcal{B}} \|f_t - f^*\|_n^2 \gg \frac{1}{\eta_{\widetilde{T}}}.$$

Combining with Theorem 4.1, we claim that $\widetilde{T}$ is a "sharp" stopping time for estimation.

At last, we comment briefly that the stopping rule for estimation and Theorem 4.1 (a), (b) can also be obtained in our framework as a by-product. Intuitively, the stopping time $\widetilde{T}$ in (4.1) is achieved by the classical bias-variance tradeoff. Note that $\|f_t - f^*\|_n^2$ has a trivial upper bound

$$\|f_t - f^*\|_n^2 \leq 2 \underbrace{\|f_t - \mathrm{E}_\epsilon f_t\|_n^2}_{\text{Variance}} + 2 \underbrace{\|\mathrm{E}_\epsilon f_t - f^*\|_n^2}_{\text{Squared bias}},$$

where the expectation is taken with respect to $\epsilon$. The squared bias term is bounded by $\frac{1}{\eta_t}$ (see Lemma A.3); the variance term is bounded by the mean of $D_{n,t}$, that is, $\|f_t - \mathrm{E} f_t\|_n^2 = O_P(\mu_{n,t})$ (see Lemma A.1), where $\mu_{n,t} = \mathrm{tr}((I - S^t)^2)/n \asymp \frac{1}{n} \sum_{i=1}^{n} \min\{1, \eta_t \widehat{\mu}_i\}$ as shown in Lemma 3.2. Obviously, according to (4.1), when $t \ll \widetilde{T}$, the squared bias will dominate the variance.

# 5 Numerical Study

In this section, we compare our testing method with an oracle version of stopping rule that uses knowledge of $f^*$, as well as the test based on the penalized regularization. We further conduct the simulation studies to verify our theoretical results.

**An oracle version of early stopping rule** The early stopping rule defined in (3.1) involves the bias of the iterated estimator $f_t$ that can be directly calculated as

$$\| \operatorname{E} f_t - f^* \|_n^2 = \| S^t U^\top f^* \|_n^2 = \frac{1}{n} \sum_{i=1}^n (S_{ii}^t)^2 [U^\top f^*(\boldsymbol{x})]_i^2.$$

And the standard derivation of $D_{n,t}$ is $\sigma_{n,t} = \frac{1}{n} \sqrt{2 \operatorname{tr} \left( (I - S^t)^4 \right)}$. An "oracle" method is to base its stopping time on the exact in-sample bias of $f_t$ and the standard derivation of $D_{n,t}$, which is defined as follows:

$$T^\dagger := \operatorname{argmin} \left\{ t \in \mathbb{N} \mid \frac{1}{n} \sum_{i=1}^n (S_{ii}^t)^2 [U^\top f^*(\boldsymbol{x})]_i^2 < \frac{1}{n} \sqrt{2 \operatorname{tr} \left( (I - S^t)^4 \right)} \right\}. \qquad (5.1)$$

Our oracle method represents an ideal case that the true function $f^*$ is known.

**Algorithm based on the early stopping rule** (3.1) In the early stopping rule defined in (3.1), the bias term is bounded by the order of $\frac{1}{\eta_t}$. To implement the stopping rule in (3.1) practically, we propose a boostrap method to approximate the bias term. Specifically, we calculate a sequence of $\{f_t^{(b)}\}_{b=1}^B$ based on the pair boostrapped data $\{x_i^{(b)}, y_i^{(b)}\}_{i=1}^n$, and use $\| S^t U^\top f_{tB} \|_n^2$ to approximate the bias term, where $f_{tB} = \frac{\sum_{b=1}^B f_{tb}}{B}$, $B$ is a positive integer. On the other hand, the standard derivation term $\frac{1}{n} \sqrt{2 \operatorname{tr} \left( (I - S^t)^4 \right)}$ involves calculating all eigenvalues of the kernel matrix. This step can be implemented by many methods on fast computation of kernel eigenvalues; see Stewart [2002], Drineas and Mahoney [2005] and reference therein.

**Penalization-based test** As another reference, we also conduct the penalization-based test by using the test statistic as $D_{n,\lambda} = \| \widehat{f}_{n,\lambda} \|_n^2$. Here $\widehat{f}_{n,\lambda}$ is the kernel ridge regression (KRR) estimator (Shawe-Taylor and Cristianini [2004]) defined as

$$\widehat{f}_{n,\lambda} := \operatorname*{argmin}_{f \in \mathcal{H}} \left\{ \frac{1}{n} \sum_{i=1}^n (y_i - f(x_i))^2 + \lambda \| f \|_\mathcal{H}^2 \right\}, \qquad (5.2)$$

where $\| f \|_\mathcal{H}^2 = \langle f, f \rangle_\mathcal{H}$ with $\langle \cdot, \cdot \rangle_\mathcal{H}$ the inner product of $\mathcal{H}$. The penalty parameter $\lambda$ plays the same role of the total step size $\eta_t$ to avoid overfitting. Liu et al. [2018] shows that minimax optimal testing rate can be achieved by choosing the penalty parameter satisfying $\lambda \asymp \sqrt{\operatorname{tr} \left( (\Lambda + \lambda I_n)^{-1}) \Lambda \right)^4} / n$. The specific $\lambda$ varies for different kernel classes. For example, in PDK, the optimal testing can be achieved with $\lambda^* \asymp n^{-4m/(4m+1)}$; in EDK, the corresponding $\lambda^* \asymp n^{-1}(\log n)^{1/(2p)}$. We discover an interesting connection that the inverse of these $\lambda^*$ share the same order as the stopping rules in Corollary 3.4 and Corollary 3.5, respectively. Lemma 5.1 provides a theoretical explanation for such connection.

**Lemma 5.1.** $\operatorname{tr} \left( (\Lambda + \lambda I_n)^{-1} \Lambda \right)^4 \asymp \operatorname{tr} \left( I - S^t \right)^4$ *holds if and only if* $\lambda \asymp \frac{1}{\eta_t}$.

However, it is still challenging to choose the optimal penalty parameter for testing in practice. A compromising strategy is to use cross validation (CV) method (Golub et al. [1979]), which was invented for optimal estimation problems. In the following numerical study, we will show that the CV-based $D_{n,\lambda}$ performs less satisfactorily than our proposed early stopping method.

## 5.1 Numerical study I

In this section, we compare our early stopping based test statistics (ES) with two other methods: the oracle early stopping (Oracle ES) method, and the penalization-based test described above. Particularity, we consider the hypothesis testing problem $H_0 : f = 0$.

Data were generated from the regression model (2.1) with $f(x_i) = c \cdot \cos(4\pi x_i)$, where $x_i \overset{iid}{\sim}$ Unif$[0, 1]$ and $c = 0, 1$ respectively. $c = 0$ is used for examining the size of the test, and $c = 1$ is used for examining the power of the test. The sample size $n$ is ranged from 100 to 1000. We use Gaussian kernel (i.e., $p = 2$ in EDK) to fit the data. Significance level was chosen as 0.05. Both size and power were calculated as the proportions of rejections based on 500 independent replications. For the ES, we use bootstrap method to approximate the bias with $B = 10$ and the step size $\alpha = 1$. For the penalization-based test, we use $10-$fold cross validation (10-fold CV) to select the penalty parameter. For the oracle ES, we follow the stopping rule in (5.1) with constant step size $\alpha = 1$.

Figure 2 (a) shows that the size of the three testing methods approach the nominal level 0.05 under various $n$, demonstrating the testing consistency. Figure 2 (b) displays the power of the three testing rules. The ES exhibits better power performance than the penalization-based test with $10-$fold CV under various sample sizes. Furthermore, as $n$ increases, the power of ES approaches to the Oracle ES, which serves as the benchmark. As shown in Figure 2 (c), the ES enjoys great computation efficiency compared with the Wald-test with $10-$fold CV, and it is reasonable that our proposed ES takes more time than the oracle ES, due to the extra step for bootstrapping. In Supplementary A.8, we show additional synthetic experiments with various $c$ based on second-order Sobolev kernel verifying our theoretical contribution.

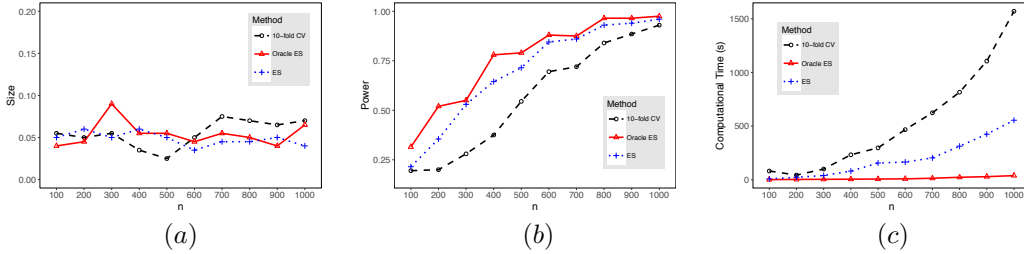

Figure 2: $(a)$ is the size with signal strength $c = 0$; $(b)$ is the power with signal strength $c = 1$; $(c)$ is the computational time (in seconds) for the three testing rules.

## 5.2  Numerical study II

In this section, we show synthetic experiments verifying our sharpness results stated in Corollary 3.4, Corollary 3.5 and Theorem 3.6. Data were generated from the regression model (2.1) with $f(x_i) = c(0.8(x_i - 0.5)^2 + 0.2 \sin(4\pi x_i))$, where $x_i \overset{iid}{\sim}$ Unif$[0, 1]$, and $c = 0, 1$, respectively. Other setting is as the same as in Section 5.1.

In Figure 3 (a) and (b), we use the second-order Sobolev kernel (i.e., $m = 2$ in PDK) to fit the model, and set the constant step size $\alpha = 1$. Corollary 3.4 suggests that optimal power can be achieved at the stopping time $T^* \asymp n^{8/9}$. To display the impact of the stopping time on power performance, we set the total iteration steps $T$ as $(n^{8/9})^\gamma$ with $\gamma = 2/3, 1, 4/3$ and $n$ ranges from 100 to 1000. Figure 3 (a) shows that the size approaches the nominal level 0.05 under various choices of $(\gamma, n)$, demonstrating the testing consistency supported by Theorem 3.1. Figure 3 (b) displays the power of our testing rule. A key observation is that the power under the theoretically derived stopping rule $(\gamma = 1)$ performs best, compared with other stopping choices $(\gamma = 2/3, 4/3)$. In Figure 3 (c) and (d), we use Gaussian kernel (i.e., $p = 2$ in EDK) to fit the model, and set the constant step size $\alpha = 1$. Here we set the total iteration steps as $(n/(\log n)^{1/4})^\gamma$ with $\gamma = 2/3, 1, 4/3$ and $n$ ranges from 100 to 1000. Note that $\gamma = 1$ corresponds to the optimal stopping time in Corollary 3.5. Overall, the interpretations are similar to Figure 3 (a) and (b) for PDK.

## 6  Discussion

The main contribution of this paper is that we apply the early stopping strategy to nonparametric testing, and propose the first "sharp" stopping rule to guarantee minimax optimal testing (to the best of our knowledge). Our stopping rule depends on the eigenvalues of the kernel matrix, especially the first few leading eigenvalues. There are many efficient methods to compute the top eigenvalues fast,

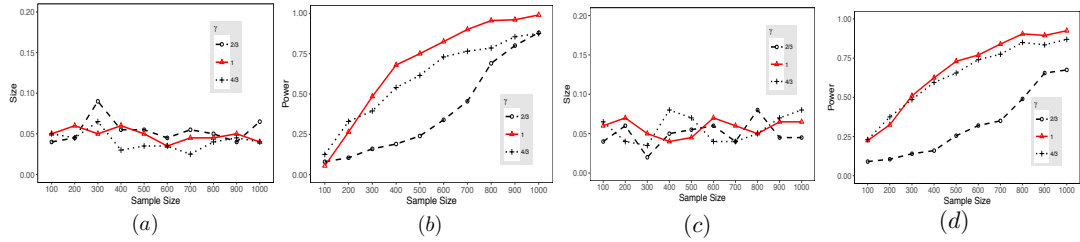

Figure 3: $(a)$ is the size of $D_{n,t}$ with signal strength $c = 0$ under PDK; $(b)$ is the power of $D_{n,t}$ with signal strength $c = 1$ under PDK. $(c)$ is the size of $D_{n,t}$ with signal strength $c = 0$ under EDK; $(d)$ is the power of $D_{n,t}$ with signal strength $c = 1$ under EDK.

see Drineas and Mahoney [2005], Ma and Belkin [2017]. As a future work, we can also introduce the randomly projected kernel methods to accelerate the computing time.

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
