[Supplementary Material]



## Early Stopping for Nonparametric Testing

# A  Proof

## A.1  Proof of Theorem 3.1

Denote $\gamma^t = \frac{1}{\sqrt{n}} U^\top \boldsymbol{f}_t$, then $\boldsymbol{f}_t = \sqrt{n} U \gamma^t$. The recursion equation of the gradient descent algorithm in (2.3) is equivalent to

$$\sqrt{n} U \gamma^{t+1} = \sqrt{n} U \gamma^t - \sqrt{n}\alpha_t \boldsymbol{K} U \gamma^t + \alpha^t \boldsymbol{K} \boldsymbol{y}. \tag{A.1}$$

Note that $\boldsymbol{y} = \boldsymbol{f}^* + \epsilon = \sqrt{n} U \gamma^* + \sqrt{n} w$, where $\boldsymbol{f}^* = (f^*(x_1), \cdots, f^*(x_n)) = \sqrt{n} U \gamma^*$, $\epsilon = (\epsilon_1, \cdots, \epsilon_n)^\top$, and $w = \frac{\epsilon}{\sqrt{n}}$. For theoretical convenience, we suppose $\sigma = 1$. Then (A.1) becomes

$$\gamma^{t+1} = \gamma^t - \alpha_t \Lambda \gamma^t + \alpha_t \Lambda \gamma^* + \alpha_t w. \tag{A.2}$$

Recall the diagonal shrinkage matrices $S^t$ at step $t$ is defined as follows

$$S^t = \prod_{\tau=0}^{t-1} \left( I_n - \alpha_\tau \Lambda \right) \in \mathbb{R}^{n \times n}.$$

Then based on (A.2), we have

$$\gamma^t - \gamma^* = \left( I - S^t \right) w - S^t \gamma^*. \tag{A.3}$$

The test statistics $D_{n,t}$ can be written as

$$D_{n,t} = \|\boldsymbol{f}_t\|_n^2 = \frac{1}{n} \boldsymbol{f}_t^\top \boldsymbol{f}_t = \gamma^{t\top} \gamma^t = \|\gamma^t\|_2^2, \tag{A.4}$$

where $\|\cdot\|_2$ is the Euclidean norm. Next, we analyze the null limiting distribution of $\|\gamma^t\|_2^2$. Under the null hypothesis, $\gamma^* = 0$, plugging (A.3) in (A.4), we have $D_{n,t} = \|\gamma^t\|_2^2 = w^\top (I_n - S^t)^2 w = \frac{1}{n}\epsilon^\top (I_n - S^t)^2 \epsilon$.

We first derive the null limiting distribution of $D_{n,t}$ conditional on $\boldsymbol{x}$. By the Gaussian assumption of $\epsilon$, we have $\mu_{n,t} \equiv \frac{1}{n} \mathrm{tr}\left( (I_n - S^t)^2 \right)$ and $\sigma_{n,t}^2 \equiv \frac{2}{n^2} \mathrm{tr}\left( (I - S^t)^4 \right)$. Define $U = \frac{D_{n,t} - \mu_{n,t}}{\sigma_{n,t}}$, then for any $k \in (-1/2, 1/2)$, we have

$$\begin{aligned}
&\log \mathrm{E}_\epsilon \left( \exp(ikU) \right) \\
&= \log \mathrm{E}_\epsilon \left( \exp(ik\epsilon^\top (I_n - S^t)^2 \epsilon/(n\sigma_{n,\lambda})) \right) - ik\mu_{n,t}/(n\sigma_{n,t}) \\
&= -\frac{1}{2} \log \det(I_n - 2ik(I_n - S^t)^2/(n\sigma_{n,t})) - ik\mu_{n,t}/(n\sigma_{n,t}) \\
&= ik \cdot \mathrm{tr}((I_n - S^t)^2)/(n\sigma_{n,t}) - k^2 \mathrm{tr}((I_n - S^t)^4)/(n^2\sigma_{n,t}^2) \\
&\quad + \mathcal{O}(k^3 \mathrm{tr}((I_n - S^t)^6)/(n^3\sigma_{n,t}^3)) - ik\mu_{n,t}/(n\sigma_{n,t}) \\
&= -k^2/2 + \mathcal{O}(k^3 \mathrm{tr}((I_n - S^t)^6)/(n^3\sigma_{n,t}^3)),
\end{aligned}$$

where $i = \sqrt{-1}$, $\mathrm{E}_\epsilon$ is the expectation with respect to $\epsilon$, and $I_n$ is $n \times n$ identity matrix. Therefore, to prove the normality of $U$, we need to show $\mathrm{tr}((I_n - S^t)^6)/(n^3\sigma_{n,t}^3) = o_P(1)$.

Note that $S^t = \prod_{\tau=0}^{t-1}(I_n - \alpha_\tau \Lambda) = \mathrm{diag}(S_{11}^t, \cdots, S_{nn}^t)$, where $S_{jj}^t = \prod_{\tau=0}^{t-1}(1 - \alpha_\tau \widehat{\mu}_j)$ for $j = 1, \cdots, n$. Then $\mathrm{tr}((I - S^t)^6) = \sum_{j=1}^n (1 - S_{jj}^t)^6$, $\mathrm{tr}((I - S^t)^4) = \sum_{j=1}^n (1 - S_{jj}^t)^4$, and

$$\frac{\mathrm{tr}((I_n - S^t)^6)}{n^3\sigma_{n,t}^3} = \frac{\mathrm{tr}((I_n - S^t)^6)}{\mathrm{tr}((I_n - S^t)^4)} \cdot \frac{1}{\sqrt{\mathrm{tr}((I_n - S^t)^4}} \leq \frac{1}{\sqrt{\mathrm{tr}((I_n - S^t)^4}},$$

where the last step is by Lemma A.2 that $(1 - S_{jj}^t) \asymp \min\{1, \eta_t \widehat{\mu}_j\} \leq 1$. Then, it is sufficient to prove $\mathrm{tr}((I_n - S^t)^4) \to \infty$ as $n \to \infty$ and $t \to \infty$.

Let $\widetilde{\kappa}_t = \operatorname{argmin}\{j : \widehat{\mu}_j \leq \frac{1}{\eta_t}\} - 1$, then

$$\operatorname{tr}((I - S^t)^4) = \sum_{j=1}^{n}(1 - S_{jj}^t)^4 \geq \frac{1}{2^4}\sum_{j=1}^{n}\big(\min\{1, \eta_t\widehat{\mu}_j\}\big)^4$$

$$= \frac{1}{2^4}\big(\widetilde{\kappa}_t + \sum_{j=\widetilde{\kappa}_t+1}^{n}(\eta_t\widehat{\mu}_j)^4\big) \geq \frac{\widetilde{\kappa}_t}{2^4}. \tag{A.5}$$

Therefore, when $n \to \infty$ and $t \to \infty$, by Assumption A2, we have $\eta_t \to \infty$; and by Assumption A3 and Lemma 3.1 in Liu et al. [2018], we have $\widetilde{\kappa}_t \to \infty$ with probability greater than $1 - e^{c\kappa_t}$, where $c$ is a constant. Then $\mathrm{E}_\epsilon(e^{ikU}) \longrightarrow e^{-\frac{k^2}{2}}$ with probability approaches 1 as $n \to \infty$ and $t \to \infty$.

We next consider $\mathrm{E}_{\boldsymbol{x}}\,\mathrm{E}_\epsilon(e^{ikU})$ by taking expectation w.r.t $\boldsymbol{x}$ on $\mathrm{E}_\epsilon(e^{ikU})$. We claim $\mathrm{E}_{\boldsymbol{x}}\,\mathrm{E}_\epsilon(e^{ikU}) \longrightarrow e^{-\frac{k^2}{2}}$ for $k \in (-\frac{1}{2}, \frac{1}{2})$. If not, there exists a subsequence of r.v $\{\boldsymbol{x}_{n_k}\}$, such that for $\forall \varepsilon > 0$, $|\mathrm{E}_{\boldsymbol{x}_{n_k}}\,\mathrm{E}_\epsilon\,e^{ikU} - e^{-\frac{k^2}{2}}| > \varepsilon$. On the other hand, since $\mathrm{E}_\epsilon\,e^{ikU(\boldsymbol{x}_{n_k})} \xrightarrow{P} e^{-\frac{k^2}{2}}$, which is bounded, there exists a sub-sub sequence $\{\boldsymbol{x}_{n_{k_l}}\}$, such that

$$\mathrm{E}_\epsilon\,e^{ikU(\boldsymbol{x}_{n_{k_l}})} \xrightarrow{a.s} e^{-\frac{k^2}{2}}.$$

Thus by dominate convergence theorem, $\mathrm{E}_{\boldsymbol{x}_{n_{k_l}}}\,\mathrm{E}_\epsilon\,e^{ikU} \longrightarrow e^{-\frac{k^2}{2}}$, which is a contradiction. Therefore, we have $U = \frac{D_{n,t} - \mu_{n,t}}{\sigma_{n,t}}$ asymptotically converges to a standard normal distribution.

## A.2   Proof of Theorem 3.3 (a)

*Proof.* Recall $\|f_t\|_n^2 = \gamma^{t\top}\gamma^t$ with $\gamma^t = (I - S^t)\epsilon/\sqrt{n} + (I - S^t)\gamma^*$. Therefore,

$$\gamma^{t\top}\gamma^t = \frac{1}{n}\epsilon^\top(I - S^t)^2\epsilon + \frac{2}{\sqrt{n}}\epsilon^\top(I - S^t)^2\gamma^* + \gamma^*(I - S^t)^2\gamma^* = W_1 + W_2 + W_3. \tag{A.6}$$

For $W_3$, since $\|f^*\|_n^2 = \|\gamma^*\|_2^2 \geq C_\varepsilon^2 d_{n,t}^2$,

$$W_3 = \|(I - S^t)\gamma^*\|_2^2 \geq \frac{1}{2}\|\gamma^*\|_2^2 - \|S^t\gamma^*\|_2^2 \geq \frac{C_\varepsilon^2}{2}(\frac{1}{\eta_t} + \sigma_{n,t}) - \frac{1}{e\eta_t} \geq \frac{C_\varepsilon^2\sigma_{n,t}}{2},$$

where $C_\varepsilon^2 \geq \frac{2}{e}$ is a constant, and the specfic requirement of $C_\varepsilon^2$ will be illustrated later.

Recall $W_2 = \frac{1}{\sqrt{n}}\epsilon^\top(I - S^t)^2\gamma^*$. Consider $a^\top(I - S^t)^2 a$, where $a = (a_1, \cdots, a_n) \in \mathbb{R}^n$ is an arbitrary vector. Then $a^\top(I - S^t)^2 a \leq \lambda_{\max}((I - S^t)^2)a^\top a \leq a^\top a$. For $W_2$, we have

$$\mathrm{E}_\epsilon\,W_2^2 = \gamma^{*\top}(I - S^t)^4\gamma^* \leq \gamma^{*\top}(I - S^t)^2\gamma^* = W_3.$$

Then

$$\mathrm{P}\left(|W_2| \geq \varepsilon^{-\frac{1}{2}}W_3^{1/2}\right) \leq \frac{\mathrm{E}_\epsilon\,W_2^2}{\varepsilon^{-1}W_3} \leq \varepsilon \tag{A.7}$$

Define $\mathcal{E}_1 = \{\frac{W_1 - \mu_{n,t}}{\sigma_{n,t}} \leq C_\varepsilon'\}$, where $C_\varepsilon'$ satisfies $\mathrm{P}(\mathcal{E}_1|\boldsymbol{x}) \geq 1 - \varepsilon$ for any $t \geq t_\varepsilon$ and $n \geq N_\varepsilon$, with probability greater than $1 - e^{-c\kappa_t}$. Also define $\mathcal{E}_2 = \{W_2 \geq -\varepsilon^{-1/2}W_3^{1/2}\}$ and $\mathcal{E}_3 = \{W_3 \geq C_\varepsilon^2\sigma_{n,t}/2\}$. Finally, with probablility greater than $1 - e^{-c\kappa_t}$,

$$\mathrm{P}_f\left(\frac{W_1 + W_2 + W_3 - \mu_{n,t}}{\sigma_{n,t}} \geq z_{1-\alpha/2}|\boldsymbol{x}\right)$$

$$\geq \mathrm{P}_f\left(\frac{W_2 + W_3}{\sigma_{n,t}} + \frac{W_1 - \mu_{n,t}}{\sigma_{n,t}} \geq z_{1-\alpha/2}, \mathcal{E}_1 \cap \mathcal{E}_2 \cap \mathcal{E}_3|\boldsymbol{x}\right)$$

$$\geq \mathrm{P}_f\left(\frac{W_3(1 - \varepsilon^{-1/2}W_3^{-1/2})}{\sigma_{n,t}} - C_\varepsilon' \geq z_{1-\alpha/2}, \mathcal{E}_1 \cap \mathcal{E}_2 \cap \mathcal{E}_3|\boldsymbol{x}\right)$$

$$\geq \mathrm{P}_f\left(C_\varepsilon(1 - \frac{1}{\sqrt{C_\varepsilon\sigma_{n,t}\varepsilon}}) - C_\varepsilon' \geq z_{1-\alpha/2}, \mathcal{E}_1 \cap \mathcal{E}_2 \cap \mathcal{E}_3|\boldsymbol{x}\right)$$

$$= \mathrm{P}_f(\mathcal{E}_1 \cap \mathcal{E}_2 \cap \mathcal{E}_3|\boldsymbol{x})$$

$$\geq 1 - 2\varepsilon$$

The second to the last equality is achieved by choosing $C_\varepsilon$ to satisfy

$$\frac{1}{\sqrt{C_\varepsilon}\sigma_{n,t}\varepsilon} < \frac{1}{2} \quad \text{and} \quad \frac{1}{2}C_\varepsilon - C'_\varepsilon \geq z_{1-\alpha/2}.$$

$\square$

### A.3 Proof of Corollary 3.4 and Corollary 3.5

We first prove Corollary 3.4.

*Proof.* By the stopping rule (3.1), at $T^*$, we have

$$\frac{1}{\eta_{T^*}} \asymp \frac{1}{n}\sqrt{\sum_{i=1}^{n}\min\{1, \eta_{T^*}\widehat{\mu}_i\}}.$$

On the other hand, suppose $T^* < n$, then with probability at least $1 - e^{-c\kappa_{T^*}}$,

$$\sum_{i=1}^{n}\min\{1, \eta_{T^*}\widehat{\mu}_i\} = \widetilde{\kappa}_{T^*} + \eta_{T^*}\sum_{i=\widetilde{\kappa}_{T^*}+1}^{n}\widehat{\mu}_i \asymp \widetilde{\kappa}_{T^*},$$

the last step is by Lemma A.4. Then we have

$$\frac{1}{\eta_{T^*}} \asymp \frac{\sqrt{\widetilde{\kappa}_{T^*}}}{n}.$$

By Lemma A.5 (a), with probability at least $1 - e^{-c_m n\kappa_{T^*}^{-4m/(2m-1)}}$, $\widetilde{\kappa}_{T^*} \asymp \kappa_{T^*}$. Then $\frac{1}{\eta_{T^*}} \asymp \frac{\sqrt{\kappa_{T^*}}}{n}$ with $\kappa_{T^*}$ satisfies $(\kappa_{T^*})^{-2m} \asymp \frac{1}{\eta_{T^*}}$. Finally we have $\eta_{T^*} \asymp n^{4m/(4m+1)}$, and $d_n^* \asymp n^{-2m/(4m+1)}$.

Corollary 3.5 can be achieved similarly.

$\square$

### A.4 Proof of Theorem 3.6

(1) We first consider the case when $t \ll T^*$.

*Proof.* Suppose the "true" function $f(\cdot) = f^*(\cdot) = \sum_{i=1}^{n}K(x_i, \cdot)w_i$, then $\boldsymbol{f}^* = (f^*(x_1), \cdots, f^*(x_n)) = n\boldsymbol{Kw}$, where $\boldsymbol{w} = (w_1, \cdots, w_n)$. Let $\boldsymbol{w} = U\boldsymbol{\alpha}$, then $\boldsymbol{f}^* = nUD\boldsymbol{\alpha}$, where $\boldsymbol{\alpha} = (\alpha_1, \cdots, \alpha_n)$. We construct $f^*(\cdot)$ with the coefficients $\{\alpha_\nu\}_{\nu=1}^{n}$ satisfies

$$\alpha_\nu^2 = \begin{cases} \frac{C}{2n(\kappa_t-1)}\mu_{g\kappa_t+k}^{-1} & \text{for } \nu = (g\kappa_t + k) \quad k = 1, 2, \cdots, \kappa_t - 1 \\ 0 & \text{otherwise} \end{cases} \tag{A.8}$$

Since $t \ll T^*$, by the definition of $\kappa_t$, we have $\kappa_t < \kappa_{T^*}$. Choose $g \geq 1$ to be an integer satisfying $(g+1)\kappa_t \leq \kappa_{T^*}$ and $n\eta_t^2\mu_{g\kappa_t}^3 \ll \kappa_t^{1/2}$. The existence of such $g$ can be verified directly based on the expression of the PDK and EDK eigenvalues.

Note that $\frac{1}{n} < \frac{1}{\eta_{T^*}} < \frac{1}{\eta_t}$, then by Lemma A.5, we have $\frac{1}{2}\mu_{g\kappa_t} \leq \widehat{\mu}_{g\kappa_t} \leq \frac{3}{2}\mu_{g\kappa_t}$ with probability approaches 1. Consider the event $\mathcal{A} = \{|\widehat{\mu}_{g\kappa_t} - \mu_{g\kappa_t}| \leq \frac{1}{2}\mu_{g\kappa_t}\}$, then $P(\mathcal{A}) \to 1$ as $n \to \infty$. Conditional on the event $\mathcal{A}$, we have

$$\|f\|_{\mathcal{H}}^2 = \|\sum_{i=1}^{n}K(x_i, \cdot)w_i\|_{\mathcal{H}}^2 = n\boldsymbol{\alpha}^\top D\boldsymbol{\alpha} = n\sum_{k=1}^{\kappa_t-1}\alpha_{g\kappa_t+k}^2\widehat{\mu}_{g\kappa_t+k} \leq C.$$

Furthermore, conditional on $\mathcal{A}$,

$$\|f\|_n^2 = n\boldsymbol{\alpha}^\top D^2\boldsymbol{\alpha} = n\sum_{k=1}^{\kappa_t-1}\alpha_{g\kappa_t+k}^2\widehat{\mu}_{g\kappa_t+k}^2 \geq \frac{C}{4}\widehat{\mu}_{(g+1)\kappa_t} \gg \widehat{\mu}_{\kappa_{T^*}} \geq \frac{1}{\eta_{T^*}} = d_n^*.$$

By (A.6), we have

$$D_{n,t} = \|f_t\|_n^2 = \frac{1}{n}\epsilon^\top (I - S^t)^2 \epsilon + \frac{2}{\sqrt{n}}\epsilon^\top (I - S^t)^2 \gamma^* + \gamma^*(I - S^t)^2 \gamma^* = W_1 + W_2 + W_3,$$

where $\gamma^* = \frac{1}{\sqrt{n}}U^\top \boldsymbol{f}^*$. Note that

$$W_3 = \gamma^*(I - S^t)^2 \gamma^* = n\sum_{i=1}^{n} \alpha_i^2 \widehat{\mu}_i^2 (1 - S_{ii}^t)^2 \leq \frac{C\eta_t^2}{\kappa_t - 1}\sum_{k=1}^{\kappa_t - 1} \widehat{\mu}_{g\kappa_t + k}^3 \leq C\eta_t^2 \widehat{\mu}_{g\kappa_t}^3,$$

where the first inequality is based on the property of shrinkage matrices $S^t$ in Lemma A.2. Conditional on the event $\mathcal{A}$, we have

$$W_3 \leq C\eta_t^2 \widehat{\mu}_{g\kappa_t}^3 \leq \frac{27C}{8}\eta_t^2 \mu_{g\kappa_t}^3 \ll \kappa_t^{1/2}/n,$$

where the last step is by the property on the integer $g$. Then we have $W_3 = o(\sigma_{n,t})$. By (A.7), we have $W_2 = W_1^{1/2}O_{P_f}(1) = o_{P_f}(\sigma_{n,t})$. Therefore,

$$\frac{D_{n,t} - \mu_{n,t}}{\sigma_{n,t}} = \frac{W_1 - \mu_{n,t}}{\sigma_{n,t}} + \frac{W_2 + W_3}{\sigma_{n,t}}$$

$$= \frac{W_1 - \mu_{n,t}}{\sigma_{n,t}} + o_{P_f}(\sigma_{n,t})$$

$$\xrightarrow{d} N(0,1).$$

Then we have, as $n \to \infty$, with probability approaches 1,

$$\inf_{f \in \mathcal{B}, \|f\|_n \geq C'd_n^*} \mathrm{P}_f\big(\phi_{n,t} = 1|\boldsymbol{x}\big) \leq \mathrm{P}_f\big(\phi_{n,t} = 1|\boldsymbol{x}\big) \to \alpha.$$

$\square$

(2) We next consider the case when $t \gg T^*$.

*Proof.* We still suppose the true function $f(\cdot) = f^*(\cdot) = \sum_{i=1}^{n} K(x_i, \cdot)w_i$, then $\boldsymbol{f}^* = n\boldsymbol{K}\boldsymbol{w}$, where $\boldsymbol{w} = (w_1, \cdots, w_n)$. Let $\boldsymbol{w} = U\boldsymbol{\alpha}$, then $\boldsymbol{f}^* = nUD\boldsymbol{\alpha}$, where $\boldsymbol{\alpha} = (\alpha_1, \cdots, \alpha_n)$. Construct the coefficients $\alpha_\nu$ satisfying

$$\alpha_\nu^2 = \begin{cases} \frac{C_1}{n}\frac{1}{\eta_{T^*}}\mu_\nu^{-2} & \text{for } \nu = 1; \\ 0 & \text{otherwise.} \end{cases} \tag{A.9}$$

Here $C_1$ is a constant independent with $n$. In the following analysis, we conditional on the event $\mathcal{A} = \big\{|\widehat{\mu}_1 - \mu_1| \leq \frac{1}{2}\mu_1\big\}$. First,

$$\|f\|_{\mathcal{H}}^2 = \|\sum_{i=1}^{n} K(x_i, \cdot)w_i\|_{\mathcal{H}}^2 = n\boldsymbol{\alpha}^\top D\boldsymbol{\alpha} = n\alpha_1^2 \widehat{\mu}_1 \leq \frac{3C_1}{2\eta_{T^*}}\mu_1^{-1} \leq C.$$

The last inequality is based on the fact that $\eta_{T^*} \to \infty$ as $n \to \infty$. Furthermore,

$$\|f\|_n^2 = n\boldsymbol{\alpha}^\top D^2 \boldsymbol{\alpha} = n\alpha_1^2 \widehat{\mu}_1^2 \geq \frac{C_1}{4\eta_{T^*}} \geq C_2 d_n^*,$$

with $C_1$ satisfying $C_1/4 \geq C_2$. By (A.6), we have

$$D_{n,t} = \|f_t\|_n^2 = \frac{1}{n}\epsilon^\top (I - S^t)^2 \epsilon + \frac{2}{\sqrt{n}}\epsilon^\top (I - S^t)^2 \gamma^* + \gamma^*(I - S^t)^2 \gamma^* = W_1 + W_2 + W_3,$$

where $\gamma^* = \frac{1}{\sqrt{n}}U^\top f^*(\boldsymbol{x})$. Note that

$$W_3 = \gamma^*(I - S^t)^2 \gamma^* = n\sum_{i=1}^{n} \alpha_i^2 \widehat{\mu}_i^2 (1 - S_{ii}^t)^2 \leq n\alpha_1^2 \widehat{\mu}_1^2 \leq \frac{9C_1}{4\eta_{T^*}}$$

$$\ll \sigma_{n,t} = \frac{1}{n}\sqrt{\sum_{i=1}^{n} \min\{1, \eta_t \widehat{\mu}_i\}},$$

then we have $W_3 = o(\sigma_{n,t})$. By (A.7), we have $W_2 = W_1^{1/2} O_{P_f}(1) = o_{P_f}(\sigma_{n,t})$. Therefore,

$$
\begin{aligned}
\frac{D_{n,t} - \mu_{n,t}}{\sigma_{n,t}} &= \frac{W_1 - \mu_{n,t}}{\sigma_{n,t}} + \frac{W_2 + W_3}{\sigma_{n,t}} \\
&= \frac{W_1 - \mu_{n,t}}{\sigma_{n,t}} + o_{P_f}(\sigma_{n,t}) \\
&\xrightarrow{d} N(0,1).
\end{aligned}
$$

Since $P(\mathcal{A}) \to 1$ as $n \to \infty$, we have, as $n \to \infty$, with probability approaches 1,

$$
\inf_{f \in \mathcal{B}, \|f\|_n \geq C' d_n^*} P_f(\phi_{n,t} = 1 | \boldsymbol{x}) \leq P_{f^*}(\phi_{n,t} = 1 | \boldsymbol{x}) \to \alpha.
$$

$\square$

## A.5 Proof of Sharpness in estimation

*Proof.* We first prove Theorem 4.2 (a) for PDK.

Suppose the true function $f(\cdot) = f^*(\cdot) = \sum_{i=1}^n K(x_i, \cdot) w_i$, then $\boldsymbol{f}^* = n\boldsymbol{K}\boldsymbol{w}$, where $\boldsymbol{w} = (w_1, \cdots, w_n)$. Let $\boldsymbol{w} = U\boldsymbol{\alpha}$, then $\boldsymbol{f}^* = nUD\boldsymbol{\alpha}$, where $\boldsymbol{\alpha} = (\alpha_1, \cdots, \alpha_n)$. Define $\breve{\kappa}_t = \operatorname{argmin}\{j : \mu_j < \frac{1}{3\eta_t}\} - 1$, and we construct $f^*$ with the coefficients $\alpha_\nu$ satisfying

$$
\alpha_\nu^2 = \begin{cases} \frac{C}{2n} \frac{1}{\breve{\kappa}_t} \mu_{\breve{\kappa}_t + k}^{-1} & \text{for } \nu = \breve{\kappa}_t + k, \ k = 1, \cdots, \breve{\kappa}_t/2; \\ 0 & \text{otherwise.} \end{cases} \tag{A.10}
$$

When $\eta_{\widetilde{T}} = n^{2m/(2m+1)}$, then $\kappa_{\widetilde{T}} = \operatorname{argmin}\{j : \mu_j < \frac{1}{\eta_{\widetilde{T}}}\} - 1 \lesssim n^{1/(2m+1)}$ by direct calculation with $\mu_i \asymp i^{-2m}$. Since $t \ll \widetilde{T}$, by Assumption A2, $\eta_t \ll \eta_{\widetilde{T}}$, then we have $\breve{\kappa}_t \leq \kappa_{\widetilde{T}} \lesssim n^{1/(2m+1)}$ and $3\breve{\kappa}_t/2 < n$.

Condition on the event $\mathcal{A} = \{|\widehat{\mu}_i - \mu_i| \leq \frac{1}{2}\mu_i\}$, it is easy to see

$$
\|f\|_{\mathcal{H}}^2 = n\alpha^\top D\alpha \leq C.
$$

Note that

$$
\begin{aligned}
\|f_t - f^*\|_n^2 &= \|\operatorname{E}_\epsilon f_t - f^*\|_n^2 + \|f_t - \operatorname{E}_\epsilon f_t\|_n^2 + \frac{2}{n}(\boldsymbol{f}_t - \operatorname{E}_\epsilon \boldsymbol{f}_t)^\top (\operatorname{E}_\epsilon \boldsymbol{f}_t - \boldsymbol{f}^*) \\
&\equiv W_1 + W_2 + W_3.
\end{aligned} \tag{A.11}
$$

Consider the bias term $W_1$, since $\boldsymbol{f}_t = \sqrt{n}U\gamma^t$ with $\gamma^t = (I - S^t)w - (I - S^t)\gamma^*$, where $\gamma^* = \sqrt{n}D\boldsymbol{\alpha}$, we have

$$
W_1 = \|\gamma^t - \gamma^*\|_2^2 = \gamma^{*\top} S^2 \gamma^* = n\boldsymbol{\alpha}^\top D^2 S^2 \boldsymbol{\alpha} = n\sum_{i=1}^n \alpha_i^2 \widehat{\mu}_i^2 S_{ii}^2
$$

By Lemma A.2, we have $S_{ii}^t \geq 1 - \min\{1, \eta_t\widehat{\mu}_i\}$. Condition on the event $\mathcal{A} = \{|\widehat{\mu}_i - \mu_i| \leq \frac{1}{2}\mu_i\}$, we have $\eta_t\widehat{\mu}_{\breve{\kappa}_t+1} \leq \frac{3}{2}\eta_t\mu_{\breve{\kappa}_t+1} \leq \frac{1}{2}$, then $0 \leq \min\{1, \eta_t\widehat{\mu}_i\} < \frac{1}{2}$ for $i = \breve{\kappa}_t + 1, \cdots, \breve{\kappa}_t + \breve{\kappa}_t/2$. Then

$$
\begin{aligned}
W_1 &\geq n\sum_{i=1}^n \alpha_i^2 \widehat{\mu}_i^2 (1 - \min\{1, \eta_t\widehat{\mu}_i\})^2 \geq \frac{n}{4} \sum_{k=1}^{\breve{\kappa}_t/2} \alpha_{\breve{\kappa}_t+k}^2 \widehat{\mu}_{\breve{\kappa}_t+k}^2 \\
&\geq \sum_{k=1}^{\breve{\kappa}_t/2} \frac{C}{8\breve{\kappa}_t} \widehat{\mu}_{\breve{\kappa}_t+k} \geq \frac{C}{16} \widehat{\mu}_{\frac{3\breve{\kappa}_t}{2}} \geq \frac{C}{32} \mu_{\frac{3\breve{\kappa}_t}{2}} \geq c_m(\breve{\kappa}_t)^{-2m} \geq c_m' \mu_{\breve{\kappa}_t} \geq \frac{c_m'}{3\eta_t},
\end{aligned}
$$

where the sixth inequality is based on the PDK's property that $\mu_i \asymp i^{-2m}$, $c_m$, $c_m'$ are constants depend on $m$.

On the other hand, by Lemma A.3, $W_1 \lesssim \frac{1}{\eta_t}$. Therefore, $W_1 = \mathcal{O}_P(\frac{1}{\eta_t})$. Furthermore, by the proof of Lemma A.1, we have $W_2 = \mathcal{O}_P(\mu_{n,t})$. By the stopping rule defined in (4.1), when $t \ll \widetilde{T}$, $\frac{1}{\eta_t} \gg \mu_{n,t}$. Then we have $W_2 = o_p(W_1)$, and $W_3 = o_P(W_1)$ due to Cauchy-Schwarz inequality $W_3 \le W_1^{1/2} W_2^{1/2}$. Finally, by Lemma A.5, with probability approaching 1,

$$\sup_{f \in \mathcal{B}} \|f_t - f^*\|_n^2 \gtrsim \sup_{f \in \mathcal{B}} \|\operatorname{E}_\epsilon f_t - f^*\|_n^2 \gtrsim \frac{1}{\eta_t} \gg \frac{1}{\eta_{\widetilde{T}}}.$$

We next prove Theorem 4.2 (b) for EDK. Similar to the proof of Theorem 4.2 (a), we construct the coefficients $\{\alpha_\nu\}_{\nu=1}^n$ as

$$\alpha_\nu^2 = \begin{cases} \frac{C}{2n} \mu_{\check{\kappa}_t+1}^{-1} & \text{for } \nu = \check{\kappa}_t + 1; \\ 0 & \text{otherwise.} \end{cases} \tag{A.12}$$

Then, it is easy to see that, conditional on $\mathcal{A}$, $\|f\|_{\mathcal{H}}^2 = n\alpha^\top D\alpha \le C$. Equation (A.11) also holds in $EDK$. $W_1 = \|\operatorname{E}_\epsilon f_t - f^*\|_n^2$ can be lower bounded as follows

$$W_1 \ge n \sum_{i=1}^n \alpha_i^2 \widehat{\mu}_i^2 (1 - \min\{1, \eta_t \widehat{\mu}_i\})^2 \ge \frac{n}{4} \alpha_{\check{\kappa}_t+1}^2 \widehat{\mu}_{\check{\kappa}_t+1}^2 \ge \frac{C}{8} \widehat{\mu}_{\check{\kappa}_t+1} \ge \frac{C}{16} \mu_{\check{\kappa}_t+1} \gg \mu_{\kappa_{\widetilde{T}}} > \frac{1}{\eta_{\widetilde{T}}},$$

where the second to last step is based on $\check{\kappa}_t + 1 \ll \kappa_{\widetilde{T}}$, which will be shown in the following. By the definition of $\check{\kappa}_t$, $\mu_{\check{\kappa}_t} > \frac{1}{3\eta_t}$, then $\check{\kappa}_t < (\frac{\log 3\eta_t}{\beta})^{1/p}$ by plugging in $\mu_i \asymp \exp(-\beta i^p)$. Similarly, $\kappa_{\widetilde{T}} > (\frac{\log \eta_{\widetilde{T}}}{\beta})^{1/p} - 1$. By Assumption A2, as $t \ll \widetilde{T}$, $\eta_t \ll \eta_{\widetilde{T}} = n/(\log n)^{1/p}$ with $n$ diverges, we have

$$\check{\kappa}_t + 1 < \left(\frac{\log 3\eta_t}{\beta}\right)^{1/p} + 1 \ll \left(\frac{\log \eta_{\widetilde{T}}}{\beta}\right)^{1/p} - 1 < \kappa_{\widetilde{T}}.$$

The analysis of $W_2$ and $W_3$ are as the same in the proof of Theorem 4.2 (a). Finally we have with probability approaching 1,

$$\sup_{f \in \mathcal{B}} \|f_t - f^*\|_n^2 \gtrsim \sup_{f \in \mathcal{B}} \|\operatorname{E}_\epsilon f_t - f^*\|_n^2 \gg \frac{1}{\eta_{\widetilde{T}}}.$$

$\square$

We provide the following lemma to bound the variance of $f_t$.

**Lemma A.1.** *Suppose Assumption A2 is satisfied. Then for $t = 1, 2, \cdots$, it holds that*
$$\|f_t - \operatorname{E}_\epsilon f_t\|_n^2 = O_P(\mu_{n,t})$$
*where $\mu_{n,t} \asymp \frac{1}{n} \sum_{i=1}^n \min\{1, \eta_t \widehat{\mu}_i\}$.*

*Proof.* First, by (A.3) and the fact that $\boldsymbol{f}_t = \sqrt{n} U\gamma^t$, we have $\operatorname{E}_\epsilon \boldsymbol{f}_t = (I_n - S^t) \boldsymbol{f}^*$. Thus the squared bias $\|\operatorname{E}_\epsilon f_t - f^*\|_n^2 = \|S^t f^*\|_n^2 = \|S^t \gamma^*\|_2^2$. By Lemma A.3, $\|\operatorname{E}_\epsilon f_t - f^*\|_n^2 \le \frac{C}{e\eta_t}$. Next, we consider the variance $\|f_t - \operatorname{E}_\epsilon f_t\|_n^2$. Note that $\|f_t - \operatorname{E}_\epsilon f_t\|_n^2 = \frac{\epsilon^\top}{\sqrt{n}} (I - S^t)^2 \frac{\epsilon}{\sqrt{n}}$, where $\|\frac{\epsilon}{\sqrt{n}}\|_{\psi_2} \le \frac{L}{\sqrt{n}}$ and $\|(I - S^t)^2\|_{op} \le 1$. Recall $\|\cdot\|_{\psi_2}$ is the sub-Gaussian norm defined as $\|\epsilon\|_\psi = \sup_{p \ge 1} p^{-1/2} (\operatorname{E}|\epsilon|^p)^{1/p}$. Here $\|\epsilon\|_{\psi_2} \le L$, with $L$ as an absolute constant. Then by Hanson-Wright concentration inequality (Rudelson and Vershynin [2013]),

$$\operatorname{P}\Big(\|f_t - \operatorname{E}_\epsilon f_t\|_n^2 - \operatorname{E}_\epsilon \|f_t - \operatorname{E}_\epsilon f_t\|_n^2 \ge \frac{\operatorname{tr}((I - S^t)^2)}{2n} \Big| \boldsymbol{x}\Big)$$
$$= \operatorname{P}\Big(\frac{1}{n} \epsilon^\top (I - S^t)^2 \epsilon - \frac{\operatorname{tr}((I - S^t)^2)}{n} \ge \frac{\operatorname{tr}((I - S^t)^2)}{2n} \Big| \boldsymbol{x}\Big)$$
$$\le \exp\Big(-c \min\Big(\frac{\operatorname{tr}^2((I - S^t)^2)}{4K^4 \|(I - S^t)^2\|_{\operatorname{F}}^2}, \frac{\operatorname{tr}((I - S^t)^2)}{\|(I - S^t)^2\|_{op}}\Big)\Big)$$
$$\le \exp(-c \operatorname{tr}((I - S^t))^2)),$$

where $\|\cdot\|_{\operatorname{F}}$ is the Frobenius norm. The last inequality holds by the fact that $\|(I - S^t)^2\|_{\operatorname{F}}^2 \le \|(I - S^t)^2\|_{op} \operatorname{tr}((I - S^t)^2)$ and $\|(I - S^t)^2\|_{op} \le 1$. Lastly, by (A.5), $\operatorname{tr}((I - S^t)^2) \ge \frac{\widetilde{\kappa}_t}{2^4}$, which goes to $+\infty$ as $t \to \infty$. Then we have, with probability approaching 1, $\|f_t - \operatorname{E}_\epsilon f_t\|_n^2 \le \frac{3}{2} \mu_{n,t}$.

$\square$

## A.6 Proof of Lemma 5.1

*Proof.* Note that $\text{tr}\left((\Lambda + \lambda I_n)^{-1}\Lambda\right)^4 \asymp \text{tr}\left(I - S^t\right)^4$ is equivalent to $\text{tr}\left((\Lambda + \lambda I_n)^{-1}\Lambda\right) \asymp \text{tr}\left(I - S^t\right)$. Let $\kappa_\lambda = \text{argmin}\{j : \widehat{\mu}_j \leq \lambda\} - 1$, then

$$\text{tr}\left((\Lambda + \lambda I_n)^{-1}\Lambda\right) = \sum_{i=1}^{\kappa_\lambda} \frac{\widehat{\mu}_i}{\widehat{\mu}_i + \lambda} + \sum_{i=\kappa_\lambda+1}^{n} \frac{\widehat{\mu}_i}{\widehat{\mu}_i + \lambda}$$

For $i \leq \kappa_\lambda$, we have $0 < \lambda < \widehat{\mu}_i$, then $\frac{1}{2}\kappa_\lambda \leq \sum_{i=1}^{\kappa_\lambda} \frac{\widehat{\mu}_i}{\widehat{\mu}_i+\lambda} \leq \kappa_\lambda$. For $i > \kappa_\lambda$, we have $0 \leq \widehat{\mu}_i < \lambda$, then $\frac{1}{2\lambda}\sum_{i=\kappa_\lambda+1}^{n} \widehat{\mu}_i \leq \sum_{i=\kappa_\lambda+1}^{n} \frac{\widehat{\mu}_i}{\widehat{\mu}_i+\lambda} \leq \frac{1}{\lambda}\sum_{i=\kappa_\lambda+1}^{n} \widehat{\mu}_i$. Therefore,

$$\text{tr}\left((\Lambda + \lambda I_n)^{-1}\Lambda\right) \asymp \kappa_\lambda + \frac{1}{\lambda}\sum_{i=\kappa_\lambda+1}^{n} \widehat{\mu}_i \asymp \sum_{i=1}^{n} \min\{1, \frac{1}{\lambda}\widehat{\mu}_i\}.$$

On the other hand, by Lemma A.2, we have $\text{tr}\left(I - S^t\right) \asymp \sum_{i=1}^{n} \min\{1, \eta_t \widehat{\mu}_i\}$. Then, it is obvious that $\text{tr}\left((\Lambda + \lambda I_n)^{-1}\Lambda\right) \asymp \text{tr}\left(I - S^t\right)$ holds if and only if $\lambda \asymp \frac{1}{\eta_t}$. $\square$

## A.7 Some auxiliary lemmas

**Lemma A.2** (Raskutti et al. [2014]Property of Shrinkage matrices $S^t$). *For all indices $j \in \{1, 2, \cdots, n\}$, the shrinkage matrices $S^t$ satisfy the bounds*

$$0 \leq (S^t)_{jj}^2 \leq \frac{1}{2e\eta_t\widehat{\mu}_j}, \quad \text{and}$$

$$\frac{1}{2}\min\{1, \eta_t\widehat{\mu}_j\} \leq 1 - S_{jj}^t \leq \min\{1, \eta_t\widehat{\mu}_j\}$$

**Lemma A.3** (Raskutti et al. [2014]Bounding the squared bias). $\|S^t\gamma^*\|_2^2 \leq \frac{C}{e\eta_t}$, *where $C$ is the constrain that $\|f\|_{\mathcal{H}} \leq C$.*

**Lemma A.4** (Liu et al. [2018]). *For $t \geq 0$, if $\eta_t < n$, then with probability at least $1 - 4e^{-\kappa_t}$, $\sum_{i=\widehat{\kappa}_t+1}^{n} \widehat{\mu}_i \leq C\kappa_t\mu_{\kappa_t}$, where $C > 0$ is an absolute constant.*

**Lemma A.5** (Liu et al. [2018]Properties of eigenvalues). $(a)$ *Suppose that $K$ has eigenvalues satisfying $\mu_i \asymp i^{-2m}$ with $m > 3/2$. Then for $i = 1, \cdots, n^{1/(2m)}$,*

$$P\left(|\widehat{\mu}_i - \mu_i| \leq \frac{1}{2}\mu_i\right) \geq 1 - e^{-c_m n i^{-4m/(2m-1)}}.$$

*where $c_m$ is an universal constant depending only on $m$.*

$(b)$ *Suppose that $K$ has eigenvalues satisfying $\mu_i \asymp \exp(-\beta i^p)$ with $\beta > 0$, $p \geq 1$. Then for $i = o(n^{1/2})$,*

$$P\left(|\widehat{\mu}_i - \mu_i| \leq \frac{1}{2}\mu_i\right) \geq 1 - e^{-c_{\beta,p}ni^{-2}},$$

*where $c_{\beta,p}$ is an universal constant depending only on $\beta$ and $p$.*
*For $i = O(n^{1/2})$, we have*

$$P\left(|\widehat{\mu}_i - \mu_i| \leq i\mu_i\right) \geq 1 - e^{-c'_{\beta,p}n},$$

*where $c'_{\beta,p}$ is an universal constant depending only on $\beta$ and $p$.*

## A.8 Additional Numerical study

In this section, we further compare our testing method (ES) with an oracle version of stopping rule (oracle ES) that uses knowledge of $f^*$, as well as the test based on the penalized regularization.

Data were generated from the regression model (2.1) with $f(x_i) = c(0.8(x_i - 0.5)^2 + 0.2\sin(4\pi x_i))$, where $x_i \overset{iid}{\sim} \text{Unif}[0, 1]$ and $c = 0, 0.5, 0.8, 1, 1.2$ respectively. $c = 0$ is used for examining the size of the test, and $c > 0$ is used for examining the power of the test. The sample size $n$ is ranged from

100 to 1000. We use the second-order Sobolev kernel with polynomial eigen-decay (i.e., $m = 2$) to fit the data. Significance level was chosen as $0.05$. Both size and power were calculated as the proportions of rejections based on $500$ independent replications. For the ES, we use boostrap method to approximate the bias with $B = 10$ and the step size $\alpha = 1$. For the penalization-based test, we use $10-$fold cross validation (10-fold CV) to select the penalty parameter. For the oracle ES, we follow the stopping rule in Section 5.1 with constant step size $\alpha = 1$. The power increases when the nonparametric signal $c$ increases for $c > 0$. Overall, the interpretations are similar to Figure 2 for EDK in Section 5.1.

Figure 4: $(a)$ is the size with signal strength $c = 0$; $(b)$ is the power with signal strength $c = 0.5$; $(c)$ is the power with $c = 0.8$; $(d)$ is the power with $c = 1.0$; $(e)$ is the power with $c = 1.2$; $(f)$ is the computational time (in seconds) for the three testing rules.