[Reviews · NeurIPS 2018]

Reviewer 1



In nonparametric inference, selecting regularization parameter via cross validation is usually suboptimal for testing. This paper studies the problem of applying an early stopping regularization method to facilitate nonparametric testing based on gradient descent algorithm in reproducing kernel Hilbert space. The authors develop a Wald-type test statistic based on the iterated estimator and derive the sharp stopping rule to guarantee minimax optimal testing for both polynomial decay kernel and exponential-polynomial decay kernel. The authors show the effectiveness of this approach by theoretical results and numerical experiments. Overall the result is novel and very interesting. The paper is well written and easy to understand.

Reviewer 2



The authors consider a non-parametric hypothesis testing procedure and derived optimal stopping rule for estimating regression function. Their proposed method ensures that their hypothesis testing procedure is minimax optimal and is sharp estimation accuracy. While the authors does not confirm the performance of their proopsed method through real data, the method would be novel and efficient for data science. My concern is that, in general, tuning parameter in the kernel is sensitive for performing simulations. So, I am interested in how robust the proposed method with respect to it. In addition, it is important to choose kernel function or to reduce variables, in practice. Are there possible to extend the proposed method to such a scenario?

Reviewer 3



Summary: This is a very interesting paper that provides an alternative approach to nonparametric testing. Instead of imposing a penalty to encourage bias-variance tradeoff, the paper proposed to do an “early stopping” to achieve that. The reason why this approach would work is that when applying a gradient ascent/descent approach, even the optimal solution will overfit the data, on the way to the optimal, there is a sweet spot where the variability has been removed a lot while the bias is still not too large (have not yet overfit the data). Overall, this is a nice and well-written paper and its idea is worth spreading in the community of machine learning and statistics. Detail comments: 1. In Theorem 3.1 and soon after it, there is a rule for testing H0 using the asymptotic distribution. However, if I understand it correctly, this rule will be asymptotically valid when both t and n goes to infinity. Will the stopping rule T^* diverges when n goes to infinity? Otherwise it may not really control the asymptotic type-1 error. 2. Although the power analysis has been done for the rejection rule, the type-1 error at t=T^* was not clear. In particular, the stopping rule T^* also depends on the data. So Theorem 3.1 may not be enough to guard the type-1 error. Is there any theoretical results about the control of type-1 error? There might be two possible ways to do that. One is to derive the asymptotic distribution at t=T^* when n goes to infinity. The other is via the optional stopping theorem. 3. Will this method be applicable to estimation problem? I guess the main challenge is to find the right time to stop the gradient ascent so the stopping has to be re-defined. Maybe the idea of Lepski’s adaptive rule could be useful in this case.